# AddSR: Accelerating Diffusion-based Blind Super-Resolution with Adversarial Diffusion Distillation

## Abstract

Blind super-resolution methods based on Stable Diffusion (SD) demonstrate impressive generative capabilities in reconstructing clear, high-resolution (HR) images with intricate details from low-resolution (LR) inputs. However, their practical applicability is often limited by poor efficiency, as they require hundreds to thousands of sampling steps. Inspired by Adversarial Diffusion Distillation (ADD), we incorporate this approach to design a highly effective and efficient blind super-resolution method. Nonetheless, two challenges arise: First, the original ADD significantly reduces result fidelity, leading to a perception-distortion imbalance. Second, SD-based methods are sensitive to the quality of the conditioning input, while LR images often have complex degradation, which further hinders effectiveness. To address these issues, we introduce a Timestep-Adaptive ADD (TA-ADD) to mitigate the perception-distortion imbalance caused by the original ADD. Furthermore, we propose a prediction-based self-refinement strategy to estimate HR, which allows for the provision of more high-frequency information without the need for additional modules. Extensive experiments show that our method, AddSR, generates superior restoration results while being significantly faster than previous SD-based state-of-the-art models (e.g., $7\times$ faster than SeeSR).

## 1 Introduction

Blind super-restoration (BSR) aims to convert low-resolution (LR) images that have undergone complex and unknown degradation into clear high-resolution (HR) versions. Differing from classical super-resolution (1; 2; 3; 4; 5), where the degradation process is singular and known, BSR is crafted to enhance real-world degraded images, imbuing them with heightened practical value.

Generative models, *e.g.* generative adversarial network (GAN) and diffusion model, have demonstrated significant superiority in BSR task to achieve realistic details. GAN-based models (6; 7; 8; 9; 10; 11) learn a mapping from the distribution of input LR images to that of HR images with adversarial training. However, when handling natural images with intricate textures, they often struggle to generate unsatisfactory visual results due to unstable adversarial objectives (9; 12).

Recently, diffusion models (DM) (13; 14) have garnered significant attention owing to their potent generative capabilities and the ability to combine information from multiple modalities. DM-based BSR methods can be roughly divided into two categories: those without Stable Diffusion (SD) prior (15; 16; 17), and those incorporating SD prior (18; 19; 20). SD prior can significantly enhance the model's ability to capture the distribution of natural images (21), thereby enabling the generated HR images with realistic details. Given the iterative refinement nature of DM, diffusion-based methods typically outperform GAN-based ones, albeit at the expense of efficiency. Hence, there's an urgent demand for BSR models that *deliver exceptional restoration quality while maintaining high efficiency* for real-world applications.

To achieve the above goal, we draw inspiration from Adversarial Diffusion Distillation (ADD) (22) and introduce it into the BSR task. However, two key challenges still exist: 1) *Perception-distortion imbalance* (23; 24; 25): Directly applying ADD in the BSR task leads to reduced fidelity, causing a perception-distortion imbalance that undermines effectiveness. 2) *Efficient restoration of high-frequency details*: The quality of the conditioning input can significantly affect the restored results

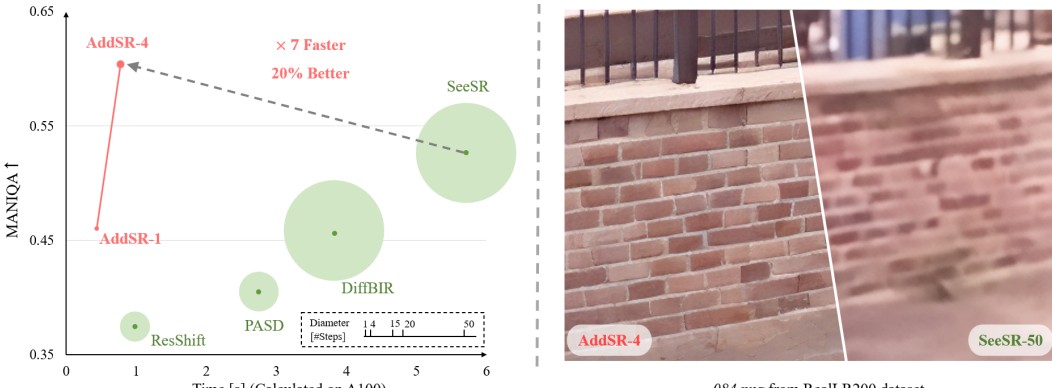

Figure 1: **Comparisons on effect and efficiency**. AddSR-4 indicates the result is obtained in 4 steps, achieving high perception quality restoration performance with the fastest speed among diffusion-based models. In contrast, existing SD-based BSR models suffer from either low perception quality restoration performance (*e.g.*, ResShift) or time-consuming efficiency (*e.g.*, SeeSR-50).

(26). Previous SD-based methods (20; 18) rely on additional degradation removal modules to pre-clean LR images for conditioning, which hinders efficiency. Therefore, efficiently obtaining a conditioning input with more high-frequency information to guide restoration is a key challenge in designing an effective and efficient blind super-resolution (BSR) method.

In this paper, we propose a novel AddSR based on ADD for blind super-restoration, which enhances restoration effects and accelerates inference speed of SD-based models simultaneously. There are two critical designs in AddSR to address the above issues respectively: 1) We introduce *timestep-adaptive adversarial diffusion distillation (TA-ADD)* loss, which designs a bivariate timestep-related weighting function to achieve perception-distortion balance, enhancing generative ability at smaller inference steps while reducing it at larger ones. 2) We propose a simple yet effective strategy, *prediction-based self-refinement (PSR)*, which uses the estimated HR image from the predicted noise to control the model output. This approach enables efficient condition restoration of the high-frequency components and further allows the restored results to contain more high-frequency details.

Our main contributions can be summarized as threefold:

• To the best of our knowledge, the proposed AddSR is the first to explore ADD for efficient and effective blind super-resolution, achieving a ×7 speedup over SeeSR(19) while delivering improved perceptual quality.

• We introduce a new TA-ADD loss to address the perception-distortion imbalance issue introduced by the original ADD, allowing AddSR to generate superior perceptual quality while maintaining comparable fidelity.

• We propose a prediction-based self-refinement (PSR) strategy to efficiently restore condition and enable the restored results to generate more details without the need for additional modules.

## 2 RELATED WORK

**GAN-based BSR.** In recent years, BSR have drawn much attention due to their practicability. Adversarial training (27; 28; 29; 30; 12) is introduced in SR task to avoid generating over-smooth results. BSRGAN (6) designs a random shuffle strategy to enlarge the degradation space for training a comprehensive SR model. Real-ESRGAN (7) presents a more practical degradation process called "high-order" to synthesize realistic LR images. KDSRGAN (8) estimates the implicit degradation representation to assist the restoration process. While GAN-based BSR methods require only one step to restore the LR image, their capability to super-resolve complex natural images is limited. In this work, our AddSR seamlessly attains superior restoration performance based on diffusion model, making it a compelling choice.

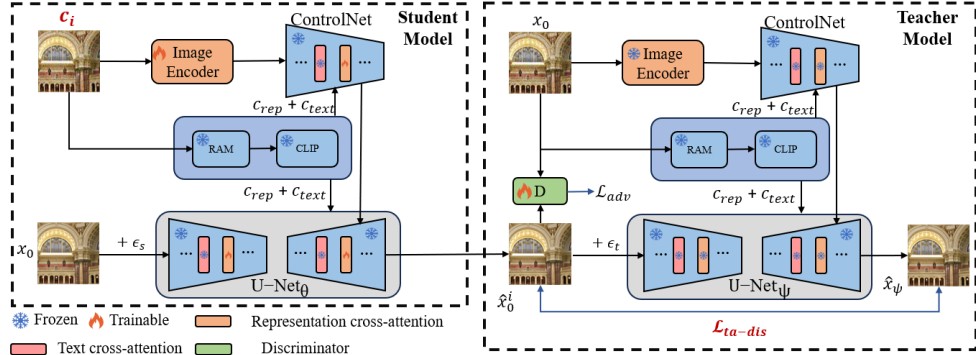

Figure 2: **Overview of AddSR**. Our proposed AddSR consists of a student model, a pretrained teacher model, and a discriminator. Let $\mathcal{C} = \{x_{LR}, \hat{x}_0^1, \hat{x}_0^2, \hat{x}_0^3\}$, where $c_i$ denotes the $i$-th element of $\mathcal{C}$ and $i$ stands for $i$-th inference step.

**Diffusion-based BSR.** Diffusion models has demonstrated significant advantages in image generation tasks (*e.g.*, text-to-image). One common approach (15; 31; 32; 33) is training a non-multimodal diffusion model from scratch, which takes the concatenation of a LR image and noise as input in every step. Another approach (21; 18; 19; 20; 34) fully leverages the prior knowledge from a pre-trained multimodal diffusion model (*i.e.*, SD model), which requires training a ControlNet and incorporates new adaptive structures (*e.g.*, cross-attention). SD-based methods excel in performance compared to the aforementioned approaches, as they effectively incorporate high-level information. However, the large number of model parameters and the need for numerous sampling steps pose substantial challenges to their application in the real world.

**Efficient Diffusion Models.** Several works (35; 14; 36; 37; 38; 39) are proposed to accelerate the inference process of DM. Although these methods can reduce the sampling steps from thousands to 20-50, the restoration effect will deteriorate dramatically. Recnet, adversarial diffusion distillation (22) is proposed to achieve 1∼4 steps inference while maintaining satisfactory generating ability. However, ADD was originally designed for the text-to-image task. Considering the multifaceted nature of BSR, such as image quality, degradation, or the trade-off between fidelity and realness, employing ADD to expedite the SD-based model for BSR is non-trivial. In contrast, AddSR introduces two pivotal designs to adapt ADD into BSR tasks, making it both effective and efficient.

## 3 METHODOLOGY

### 3.1 OVERVIEW OF ADDSR

**Network Components.** The AddSR training procedure primarily consists of three components: the student model with weights $\theta$, the pretrained teacher model with frozen weights $\psi$ and the discriminator with weights $\phi$, as depicted in Fig. 2. Specifically, both the student model and the teacher model share identical structures, with the student model initialized from the teacher model. The student model incorporates a ControlNet (40) to receive $x_{LR}$ or predicted $\hat{x}_0^{i-1}$ for controlling the output of the U-Net (41). Furthermore, the student model utilizes RAM (42) to obtain representation embeddings $c_{rep}$, extracting high-level information (*i.e.*, image content) and sends this information to CLIP (43) to generate text embeddings $c_{text}$. These embeddings help the backbone (U-Net and ControlNet) produce high-quality restored images. As for the discriminator, we adopt the same structure as StyleGAN-T (44) conditioned on $c_{img}$ extracted from $x_{LR}$ by DINOv2 (45).

**Training Procedure**. (1) **Student model with prediction-based self-refinement**. Firstly, we uniformly choose a student timestep $s$ from $\{s_1, s_2, s_3, s_4\}$ (evenly selected from 0 to 999) and employ the forward process on the HR image $x_0$ to generate the noisy state $x_s = \sqrt{\overline{\alpha}_s}x_0 + \sqrt{1 - \overline{\alpha}_s}\epsilon$. Secondly, we input $x_s$ with the condition $c_i$, the $i$-th element of $\mathcal{C} = \{x_{LR}, \hat{x}_0^1, \hat{x}_0^2, \hat{x}_0^3\}$ ($\hat{x}_0^{i-1}$ is obtained by PSR to reduce the degradation impact and provide more high-frequency information to the restoration process, as detailed in Sec. 3.2), along with $c_{rep}$ and $c_{text}$, into the student model to generate samples $\hat{x}_0^i(x_s, s, c_{rep}, c_{text}, c_i)$. (2) **Teacher model.** Firstly, we equally choose a teacher

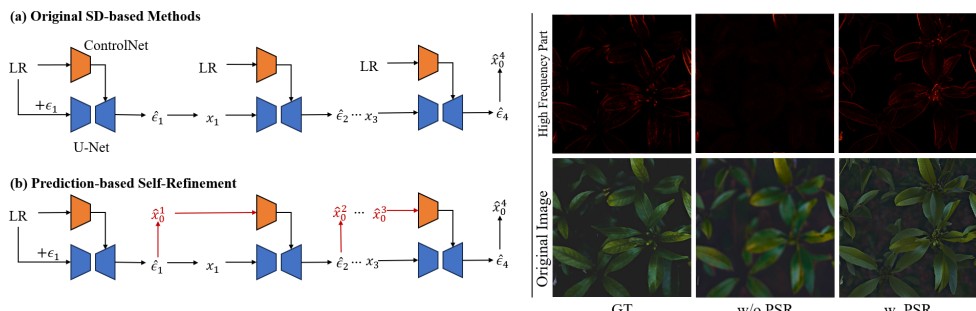

Figure 3: **Illustration of the proposed PSR**. The previous SD-based methods usually use LR image to guide model's output, while our PSR additionally utilizes the predicted HR image from the previous step to provide better supervision with marginal additional time cost.

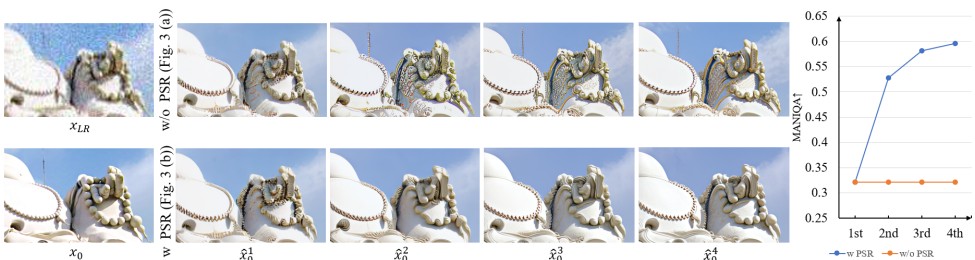

Figure 4: **Left**: Visual comparisons with and without PSR. **Right**: Perception quality of the control signal at each timestep. MANIQA is calculated between the input of ControlNet and $x_0$.

timestep $t$ from $\{t_1, t_2, ..., t_{1000}\}$ and employ forward process to student-generated samples $\hat{x}_\theta$ to obtain the noisy state $\hat{x}_{\theta,t} = \sqrt{\overline{\alpha}_t}\hat{x}_0^i + \sqrt{1-\overline{\alpha}_t}\epsilon$. Secondly, we put $\hat{x}_{\theta,t}$ with condition $x_0$, $c'_{rep}$ and $c'_{text}$ into teacher model to generate samples $\hat{x}_\psi(\hat{x}_{\theta,t}, t, c'_{rep}, c'_{text}, x_0)$. Note that $\hat{x}_\psi$ *is conditioned on $x_0$ instead of $x_{LR}$*. The primary reason is that substituting $x_{LR}$ with $x_0$ to regulate the output of the teacher model can force student model implicitly learning the high-frequency information of HR images even conditioned on $c_i$. (3) **Timestep-adaptive ADD** for BSR task. It consists of two parts: adversarial loss and a novel timestep-adaptive distillation loss, which is correlated with both the teacher and student model timesteps. The overall objective is:

$$\mathcal{L}_{TA-ADD} = \mathcal{L}_{ta-dis}(\hat{x}_0^i(x_s, s, \rho, c_i), \hat{x}_\psi(\hat{x}_{\theta,t}, t, \rho', x_0), d(s,t)) + \\ \lambda \mathcal{L}_{adv}(\hat{x}_0^i(x_s, s, \rho, c_i), x_0, \psi_{c_{img}}), \tag{1}$$

where $\rho$ denotes the $c_{rep}$ and $c_{text}$, $\rho'$ stands for $c'_{rep}$ and $c'_{text}$. $\lambda$ is the balance weight, empirically set to 0.02. $\psi_{c_{img}}$ is the discriminator conditioned on $c_{img}$. $d(s,t)$ is a weighting function defined by student timestep $s$ and teacher timestep $t$, dynamically adjusting $\mathcal{L}_{ta-dis}$ and $\mathcal{L}_{adv}$ to alleviate perception-distortion imbalance. Further analysis is provided in Sec. 3.3.

### 3.2 Prediction-based Self-Refinement

**Motivation.** As shown in Fig. 3 (a), original SD-based methods directly use LR images to control the output of DM in each inference step. However, some studies (18; 20; 26) have found that the restored results can be affected by the condition quality, as LR images often suffer from multiple degradations, which can significantly disrupt the restoration process (*e.g.*, see the first line of Fig. 4). To provide a better condition, these methods employ *additional degradation removal models* to pre-clean LR images, aiming to mitigate the impact of degradation. However, such approaches often compromise efficiency, which hinders designing an efficient method.

**Approach.** To achieve efficient restoration of high-frequency details, we propose a novel prediction-based self-refinement strategy, which incurs only minimal efficiency overhead. The core idea of PSR is to utilize the predicted noise to estimate HR. Specifically, we use the following equation:

$$\hat{x}_0 = (x_s - \sqrt{1-\overline{\alpha}_s}\epsilon_{\theta,s})/\sqrt{\overline{\alpha}_s} \tag{2}$$

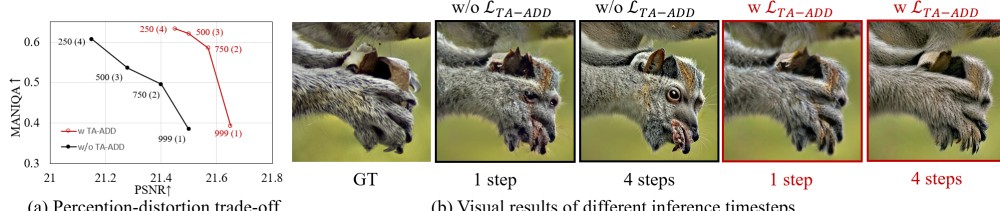

(a) Perception-distortion trade-off.    (b) Visual results of different inference timesteps.

Figure 5: **Illustrations of TA-ADD in perception-distortion trade-off**. (a) The perception and fidelity variation trends with and without TA-ADD. MANIQA and PSNR stand for perception quality and fidelity, respectively. (b) The outputs at 1st and 4th timesteps. The final output without $\mathcal{L}_{TA-ADD}$ hallucinates the paw into an animal head, while AddSR retains the appearance of the paw.

to estimate the HR image $\hat{x}_0$ from predicted noise in each step, and then control the model output in next step, where $x_s$ is the noisy state and $\epsilon_{\theta,s}$ is the predicted noise at timestep $s$. The $\hat{x}_0$ in each step has more high-frequency information to better control the model output (*e.g.*, Fig. 3-right and also Fig. 4-left). Moreover, although PSR does not use additional modules to pre-clean LR image, the HR image estimated by PSR exhibit superior quality compared to the LR image (Fig. 4-right). By leveraging our simple yet effective PSR, AddSR captures conditions with more high-frequency information, generating restored results with enhanced detail, without sacrificing efficiency.

### 3.3 TIMESTEP-ADAPTIVE ADD

**Motivation.** Perception-distortion trade-off (23) is a well-known phenomenon in SR task. We observe that training BSR task with ADD directly exacerbates this phenomenon, as shown in Fig. 5(a). Specifically, during the first three inference steps, there is a significant decrease in fidelity, accompanied by improvement in perception quality. In the last inference step, fidelity remains at a low level, while perception quality undergoes a dramatic increase. The aforementioned scenario may give rise to two issues: *(1) When the inference step is small, the quality of restored image is subpar. (2) As the inference step increases, the generated images may exhibit "hallucinations".*

**Analysis.** The primary reason lies with ADD, which maintains a consistent weight for GAN loss and distillation loss across various student timesteps, as depicted in Fig. 6(a). Once the teacher timestep is established, the ratio of adversarial loss and distillation loss remains constant for different student timesteps. However, since the perception quality of generated images gradually increases with larger inference steps, the weight-invariant ADD may lead to insufficient adversarial constraints on the student model during small inference steps, resulting in the generation of blurry images. Conversely, as the inference step increases, the adversarial training constraints become too strong, leading to the generation of "hallucinations" (see Fig. 5(b)).

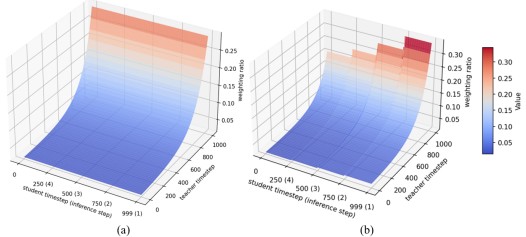

(a)      (b)

Figure 6: **Relation between weighting ratio and timesteps**. (a) weighting ratio $= \lambda / (\prod_{i=0}^{t}(1-\beta_t))^{\frac{1}{2}}$. Once the teacher timestep is established, the weighting ratio remains constant. (b) weighting ratio $= \lambda / d(s,t)$. Even the teacher timestep is established, the weighting ratio can change to balance perception-distortion across different student timesteps.

**Approach.** To address this issue, we extend the original unary weighting function $(\prod_{i=0}^{t}(1-\beta_t))^{\frac{1}{2}}$ to a bivariate weighting function $d(s,t)$, allowing for dynamic adjustment of the ratio between adversarial loss and distillation loss based on both student timestep and teacher timestep, as shown in Fig. 6(b). Specifically, we increase this ratio when only one inference step is performed, and gradually decrease it as the inference step increases. This alleviates the aforementioned issue of generating blurry images with small inference step and "hallucinations" with larger inference steps. We employ the exponential forms to control the weighting ratio. The function $d(s,t)$ can be defined as follows:

$$d(s,t) = (\prod_{i=0}^{t}(1-\beta_t))^{\frac{1}{2}} \times \mu \cdot \nu^{p(s)-1}, \tag{3}$$

Table 1: Quantitative comparison with SotAs on different degradation cases. '*' indicates that the metric is non-reference. The best results are marked in red, while the second best ones are in blue.

| Datasets | Metrics | BSRGAN ICCV 2021 | Real-ESRGAN ICCVW 2021 | MM-RealSR ECCV 2022 | LDL CVPR 2022 | FeMaSR MM 2022 | StableSR-200 Arxiv 2023 | ResShift-15 NeurIPS 2023 | PASD-20 Arxiv 2023 | DiffBIR-50 Arxiv 2023 | SeeSR-50 CVPR 2024 | AddSR-1 - | AddSR-4 - |
|---|---|---|---|---|---|---|---|---|---|---|---|---|---|
| Single: SR(×4) | MANIQA* ↑ | 0.3990 | 0.3859 | 0.3959 | 0.3501 | 0.4603 | 0.4088 | 0.4582 | 0.4405 | 0.4680 | 0.5082 | 0.3894 | 0.6430 |
| | MUSIQ* ↑ | 66.06 | 63.32 | 64.22 | 61.10 | 65.31 | 65.46 | 65.50 | 66.80 | 67.61 | 68.88 | 63.05 | 71.43 |
| | CLIPIQA* ↑ | 0.5951 | 0.5367 | 0.5967 | 0.5120 | 0.6773 | 0.6483 | 0.6803 | 0.6396 | 0.6934 | 0.7039 | 0.5572 | 0.7794 |
| | NIQE* ↓ | 5.01 | 5.21 | 5.22 | 5.39 | 5.80 | 5.35 | 5.74 | 4.68 | 4.88 | 5.06 | 5.31 | 4.75 |
| | LPIPS* ↓ | 0.2003 | 0.1962 | 0.1934 | 0.1892 | 0.1770 | 0.1944 | 0.1544 | 0.1891 | 0.2388 | 0.3085 | 0.2872 | 0.2812 |
| | PSNR↑ | 25.52 | 25.30 | 24.35 | 25.09 | 23.74 | 24.45 | 25.53 | 25.15 | 23.43 | 24.61 | 22.70 | 21.83 |
| | SSIM↑ | 0.7091 | 0.7158 | 0.7232 | 0.7282 | 0.6788 | 0.6904 | 0.7206 | 0.6896 | 0.6025 | 0.6709 | 0.6012 | 0.5651 |
| Mixture: Blur($\sigma$=2)+ SR(×4) | MANIQA* ↑ | 0.3823 | 0.3688 | 0.3796 | 0.3337 | 0.4184 | 0.3587 | 0.4195 | 0.4124 | 0.4648 | 0.4974 | 0.3779 | 0.6340 |
| | MUSIQ* ↑ | 64.73 | 60.89 | 62.21 | 58.64 | 62.96 | 60.85 | 62.02 | 64.41 | 67.09 | 68.27 | 61.95 | 71.11 |
| | CLIPIQA* ↑ | 0.5752 | 0.5116 | 0.5687 | 0.4910 | 0.6390 | 0.5819 | 0.6375 | 0.6026 | 0.6857 | 0.6892 | 0.5389 | 0.7727 |
| | NIQE* ↓ | 5.17 | 5.54 | 5.66 | 5.72 | 5.62 | 5.96 | 6.25 | 5.01 | 5.18 | 5.32 | 5.81 | 6.11 |
| | LPIPS* ↓ | 0.2240 | 0.2267 | 0.2295 | 0.2226 | 0.1979 | 0.2384 | 0.2029 | 0.2522 | 0.2522 | 0.2124 | 0.3007 | 0.2953 |
| | PSNR↑ | 25.07 | 24.74 | 24.20 | 24.45 | 24.00 | 24.01 | 24.95 | 24.70 | 22.97 | 24.12 | 22.57 | 21.69 |
| | SSIM↑ | 0.6820 | 0.6890 | 0.6927 | 0.6973 | 0.6730 | 0.6596 | 0.6926 | 0.6688 | 0.5802 | 0.6508 | 0.5905 | 0.5556 |
| Mixture: SR(×4)+ Noise($\sigma$=40) | MANIQA* ↑ | 0.2645 | 0.3120 | 0.3285 | 0.3138 | 0.3123 | 0.3485 | 0.3741 | 0.4270 | 0.4121 | 0.5537 | 0.4320 | 0.6517 |
| | MUSIQ* ↑ | 50.47 | 53.43 | 56.53 | 53.30 | 56.55 | 52.24 | 60.99 | 62.10 | 61.85 | 70.32 | 65.54 | 71.26 |
| | CLIPIQA* ↑ | 04543 | 0.4761 | 0.5158 | 0.6208 | 0.5178 | 0.4414 | 0.5949 | 0.5503 | 0.6149 | 0.7557 | 0.6219 | 0.7768 |
| | NIQE* ↓ | 7.04 | 6.00 | 4.40 | 5.61 | 4.27 | 5.12 | 6.10 | 5.02 | 5.09 | 4.95 | 4.87 | 6.29 |
| | LPIPS* ↓ | 0.4611 | 0.3601 | 0.3052 | 0.3138 | 0.3267 | 0.4017 | 0.3129 | 0.3451 | 0.3404 | 0.2999 | 0.3546 | 0.3488 |
| | PSNR↑ | 17.90 | 21.97 | 22.04 | 22.68 | 21.84 | 21.20 | 22.78 | 22.12 | 22.22 | 21.04 | 21.01 | 20.79 |
| | SSIM↑ | 0.5210 | 0.6044 | 0.5998 | 0.5838 | 0.5421 | 0.5077 | 0.5979 | 0.5587 | 0.5311 | 0.5388 | 0.5684 | 0.5621 |
| Mixture: Blur($\sigma$=2)+ SR(×4)+ Noise($\sigma$=20)+ JPEG(q=50) | MANIQA* ↑ | 0.3524 | 0.3374 | 0.3287 | 0.3082 | 0.3271 | 0.3452 | 0.3702 | 0.4024 | 0.4538 | 0.5266 | 0.3930 | 0.6335 |
| | MUSIQ* ↑ | 59.83 | 55.54 | 55.30 | 52.79 | 60.87 | 61.21 | 56.99 | 63.25 | 64.50 | 69.08 | 62.69 | 70.59 |
| | CLIPIQA* ↑ | 0.5380 | 0.5047 | 0.4978 | 0.4699 | 0.6061 | 0.6010 | 0.5888 | 0.5733 | 0.6626 | 0.7180 | 0.5669 | 0.7703 |
| | NIQE* ↓ | 5.31 | 5.69 | 5.70 | 5.77 | 4.87 | 6.33 | 7.03 | 5.49 | 4.93 | 5.06 | 5.13 | 4.68 |
| | LPIPS* ↓ | 0.3223 | 0.3346 | 0.3372 | 0.3272 | 0.2922 | 0.3429 | 0.3526 | 0.3482 | 0.3502 | 0.3085 | 0.3398 | 0.3368 |
| | PSNR↑ | 23.04 | 22.70 | 22.47 | 22.36 | 22.17 | 22.39 | 22.36 | 22.25 | 21.46 | 21.86 | 21.65 | 21.45 |
| | SSIM↑ | 0.5866 | 0.5935 | 0.5950 | 0.5948 | 0.5633 | 0.5704 | 0.5574 | 0.5594 | 0.5029 | 0.5474 | 0.5312 | 0.5210 |

where $\beta$ represents the noise schedule coefficient, with $t$ and $s$ denoting the teacher timestep and student timestep, respectively. The hyper-parameter $\mu$ sets the initial weighting ratio, while $\nu$ controls the distillation loss increase over student timesteps, typically resulting in higher fidelity with larger $\nu$. The function $p(\cdot)$ serves as a projection function that maps student timesteps to inference steps (e.g., mapping $s = 999$ to 1). We primarily consider the exponential and linear forms to control the weighting ratio. A comparison of preferences and detailed settings for different hyper-parameters are provided in Appendix Sec. C. From these comparisons, we find that the exponential form of $d(s, t)$ yields good results, so we use Eq. (3) as the distillation loss function for the remaining experiments.

# 4 EXPERIMENTS

## 4.1 EXPERIMENTAL SETTINGS

**Training Datasets.** We adopt DIV2K (46), Flickr2K (47), first 20K images from LSDIR (48) and first 10K face images from FFHQ (49) for training. We use the same degradation model as Real-ESRGAN (7) to synthesize HR-LR pairs.

**Test Datasets.** We evaluate AddSR on 4 datasets: DIV2K-val (46), DRealSR (50), RealSR (51) and RealLR200 (19). We conduct 4 degradation types on DIV2K-val to comprehensively assess AddSR, and except RealLR200, all datasets are cropped to $512 \times 512$ and degraded to $128 \times 128$ LR image.

**Implementation Details.** We adopt SeeSR (19) as the teacher model. Note that our approach is applicable to most of the existing SD-based BSR methods for improving restoration results and acceleration. The student model is initialized from the teacher model, and fine-tuned with Adam optimizer for 50K iterations. The batch size and learning rate are set to 6 and $2 \times 10^{-5}$, respectively. AddSR is trained under $512 \times 512$ resolution images with 4 NVIDIA A100 GPUs (40G).

**Evaluation Metrics.** We employ non-reference metrics (*i.e.*, MANIQA (52), MUSIQ (53), CLIP-IQA (54)) and reference metrics (*i.e.*, LPIPS (55), PSNR, SSIM (56)) to comprehensively evaluate AddSR. Non-reference metrics are prioritized as they closely align with human perception.

**Compared Methods.** Extensive state-of-the-art BSR methods are compared, including GAN-based methods: BSRGAN (6), Real-ESRGAN (7), MM-RealSR (57), LDL (9), FeMaSR (10) and diffusion-based methods: StableSR (20), ResShift (15), PASD (21), DiffBIR (18), SeeSR (19).

## 4.2 EVALUATION ON SYNTHETIC DATA

To demonstrate the superiority of the proposed AddSR in handling various degradation cases, we synthesized 4 test datasets using the DIV2K-val dataset with different degradation processes. The quantitative results are summarized in Tab. 1. Since SD-based methods emphasize perceptual quality, we provide results using perceptual-priority parameters. In the ablation study (Tab. 7), we present

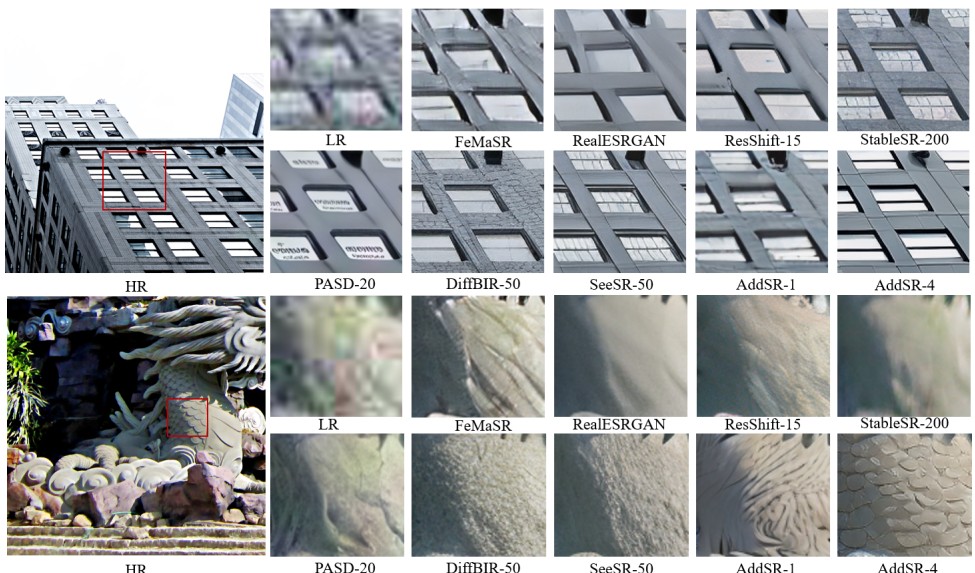

Figure 8: **Visual comparisons on synthetic LR images**. Please zoom in for a better view.

Table 2: Quantitative comparison of SUPIR: Inference time, model size, training source, and metrics.

| Model | Params [B] | Time [s] | Training Source | Dataset Size [M] | PSNR↑ | SSIM↑ | MANIQA↑ | CLIPIQA↑ |
|---|---|---|---|---|---|---|---|---|
| SUPIR (CVPR 2024) | ∼15.56 | 14.17 | 64 A6000 (48G) | 20 | 20.78 | 0.4587 | **0.6787** | **0.7992** |
| AddSR (Ours) | **∼2.28** | **0.80** | **4 A100 (40G)** | **0.034** | **21.45** | **0.5210** | 0.6335 | 0.7703 |

the corresponding results under balanced parameters. The conclusions include: (1) Our AddSR-4 achieves the highest scores in MANIQA, MUSIQ and CLIPIQA across 4 degradation cases. Especially for MANIQA, AddSR surpasses the second-best method by more than 16% on average. (2) Diffusion-based models usually achieve low scores in full-reference metrics like PSNR, SSIM and LPIPS, possibly because of their powful generative ability for realistic details that do not exist in GT.

However, full-reference metrics cannot precisely reflect human preferences (see Fig. 7), as discussed in previous works (58; 59; 60). (3) AddSR-1 can generate comparative results against other SD-based methods except SeeSR, but significantly reduces the sampling steps (*i.e.*, from ≥15 steps to only 1 step).

Moreover, we provide the comparison with SOTA perceptual method SUPIR (58) in Tab. 2, which details parameters, inference time, training sources, training data, and metrics. As shown, SUPIR exhibits better perceptual quality compared to AddSR. However, AddSR strikes a better balance among model size, inference time, fidelity, perceptual quality and training resource consumption.

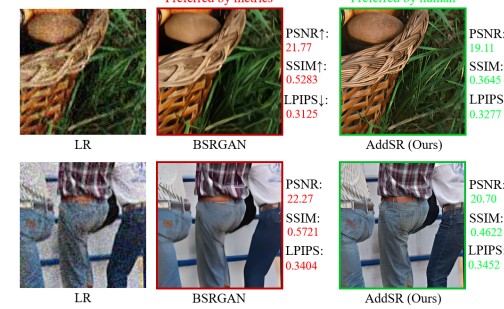

Figure 7: **Illustration on disparity** between full-reference metrics and human preference. Despite AddSR achieves lower scores in full-reference metrics, it generates human-preferred images.

For a more intuitive comparison, we provide visual results in Fig. 8. One can see that GAN-based method like FeMaSR fails to reconstruct the clean and detailed HR images of the three displayed LR images. As for SD-based method DiffBIR, it tends to generate wrong texture. This is mainly because DiffBIR uses a degradation removal structure to remove the degradation of LR images. However, the processed LR image is blurry, which may lead to the blurry results. Thanks to our proposed PSR, AddSR uses the predicted $\hat{x}_0^{i-1}$ to control the model output, which has more high-frequency information and nearly no extra time cost. With TA-ADD, AddSR can generate precise images and rich details. In a nutshell, AddSR can produce images with better perceptual quality than the state-of-the-art models while requiring fewer inference steps and less time.

Table 3: Quantitative comparison with state of the arts on real-world LR images.

| Datasets | Metrics | BSRGAN ICCV 2021 | Real-ESRGAN ICCVW 2021 | MM-RealSR ECCV 2022 | LDL CVPR 2022 | FeMaSR MM 2022 | StableSR-200 Arxiv 2023 | ResShift-15 NeurIPS 2023 | PASD-20 Arxiv 2023 | DiffBIR-50 Arxiv 2023 | SeeSR-50 CVPR 2024 | AddSR-1 - | AddSR-4 - |
|---|---|---|---|---|---|---|---|---|---|---|---|---|---|
| RealSR | MANIQA* ↑ | 0.3762 | 0.3727 | 0.3966 | 0.3417 | 0.3609 | 0.3656 | 0.3750 | 0.4041 | 0.4392 | 0.5396 | 0.4189 | 0.6597 |
| | MUSIQ* ↑ | 63.28 | 60.36 | 62.94 | 58.04 | 59.06 | 61.11 | 56.06 | 62.92 | 64.04 | 69.82 | 63.56 | 72.25 |
| | CLIPIQA* ↑ | 0.5116 | 0.4492 | 0.5281 | 0.4295 | 0.5408 | 0.5277 | 0.5421 | 0.5187 | 0.6491 | 0.6700 | 0.4929 | 0.7215 |
| | PSNR↑ | 26.49 | 25.78 | 23.69 | 25.09 | 25.17 | 25.63 | 26.34 | 26.67 | 25.06 | 25.24 | 24.22 | 22.73 |
| | SSIM↑ | 0.7667 | 0.7621 | 0.7470 | 0.7642 | 0.7359 | 0.7483 | 0.7352 | 0.7577 | 0.6664 | 0.7204 | 0.6863 | 0.6336 |
| DrealSR | MANIQA* ↑ | 0.3431 | 0.3428 | 0.3625 | 0.3237 | 0.3178 | 0.3222 | 0.3284 | 0.3874 | 0.4646 | 0.5125 | 0.3873 | 0.6034 |
| | MUSIQ* ↑ | 57.17 | 54.27 | 56.71 | 52.38 | 53.70 | 52.28 | 50.14 | 55.33 | 60.40 | 65.08 | 57.42 | 68.16 |
| | CLIPIQA* ↑ | 0.5094 | 0.4514 | 0.5171 | 0.4410 | 0.5639 | 0.5101 | 0.5287 | 0.5384 | 0.6397 | 0.6910 | 0.5543 | 0.7381 |
| | PSNR↑ | 28.68 | 28.57 | 26.84 | 27.41 | 26.83 | 29.14 | 28.27 | 29.06 | 26.56 | 28.09 | 27.49 | 26.09 |
| | SSIM↑ | 0.8022 | 0.8042 | 0.7959 | 0.8069 | 0.7545 | 0.8040 | 0.7542 | 0.7906 | 0.6436 | 0.7664 | 0.7588 | 0.7036 |
| RealLR200 | MANIQA* ↑ | 0.3688 | 0.3656 | 0.3879 | 0.3266 | 0.4099 | 0.3672 | 0.4182 | 0.4193 | 0.4626 | 0.4911 | 0.4215 | 0.6182 |
| | MUSIQ* ↑ | 64.87 | 62.93 | 65.24 | 60.95 | 64.24 | 62.89 | 60.25 | 66.35 | 66.84 | 68.63 | 65.02 | 72.62 |
| | CLIPIQA* ↑ | 0.5699 | 0.5423 | 0.6010 | 0.5088 | 0.6547 | 0.5916 | 0.6468 | 0.6203 | 0.6965 | 0.6617 | 0.5679 | 0.7724 |

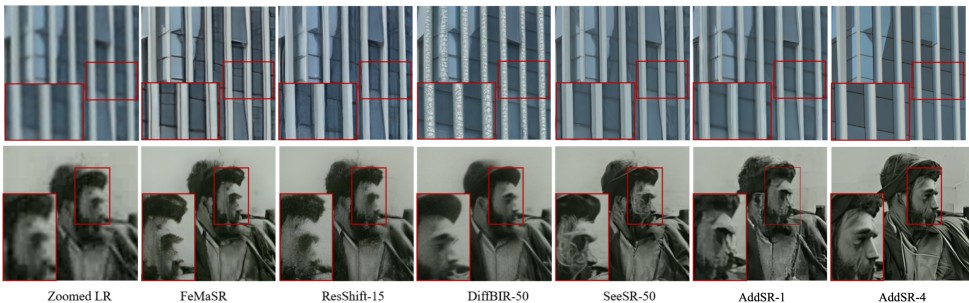

Zoomed LR    FeMaSR    ResShift-15    DiffBIR-50    SeeSR-50    AddSR-1    AddSR-4

Figure 9: **Visual comparisons on real-world LR images**. Please zoom in for a better view.

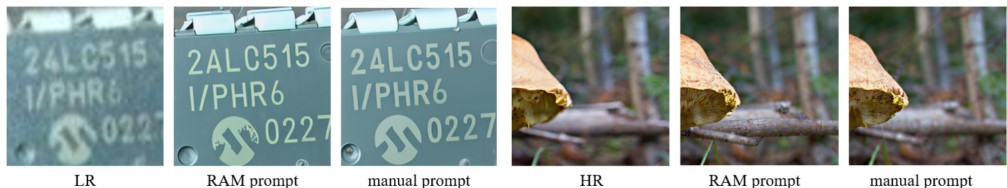

LR    RAM prompt    manual prompt    HR    RAM prompt    manual prompt

Figure 10: **Illustrations of prompt-guided restoration** that engages with manual prompts for more precise outcomes. In each group, the prompts for the second and third images are obtained through RAM and manual input, respectively. **Left**: '2ALC515' is corrected as '24LC515' by manual prompt. **Right**: The background near mushroom is modified to be blurry, aligning with GT.

### 4.3 EVALUATION ON REAL-WORLD DATA

Tab. 3 shows the quantitative results on 3 real-world datasets. We can see that our AddSR achieves the best scores in MANIQA, MUSIQ and CLIPIQA, the same as in the synthetic degradation cases. This demonstrates that AddSR has an excellent generalization ability to handle unknown complex degradations, making it practical in real-world scenarios. Additionally, AddSR-1 surpasses the GAN-based methods, primarily due to the integration of diffusion model with adversarial training. This integration enables AddSR to leverage high-level information to enhance the restoration process and generate high perception quality images, even through a *one-step* inference.

Fig. 9 shows the visualization results. We present the examples of building and face to comprehensively compare various methods. A noticeable observation is that AddSR generate more clear and regular line, as evidenced by the linear pattern of the building in the first example. In the second example, the original LR image is heavily degraded, FeMaSR and ResShift fail to generate the human face, showing only the blurry outline of the face. DiffBIR can generate more details, yet still unclear. The image generated by SeeSR exhibits artifacts. Conversely, our AddSR can generate comparative results with FeMaSR and ResShift in one-step. As evaluating the inference steps, AddSR generates more clear and detailed human face, which significantly surpasses the aforementioned methods.

**Prompt-Guided Restoration.** One of the advantages of diffusion model is to integrate with text. In Fig. 10, we demonstrate that our AddSR can efficiently achieve more precise restoration results *in 4 steps* by incorporating with manual prompt, *i.e.*, we can manually input the text description of the LR image to assist the restoration process. Specifically, in Fig. 10(a), the word on the chip can be corrected from '2ALC515' to '24LC515' with the manual prompt. In Fig. 10(b), the mushroom's background

Table 4: Quantitative comparison of SeeSR-Turbo and AddSR: Results from 2 steps.

| Methods | RealLR200 | | | | DIV2K | | | |
|---|---|---|---|---|---|---|---|---|
| | NIQE↓ | MANIQA↑ | MUSIQ↑ | CLIPIQA↑ | PSNR↑ | SSIM↑ | MANIQA↑ | CLIPIQA↑ |
| SeeSR-Turbo-2 | 7.87 | 0.3503 | 53.88 | 0.4634 | 18.45 | 0.2851 | 0.5719 | 0.6479 |
| AddSR-2 | **5.08** | **0.6182** | **72.62** | **0.7724** | **21.92** | **0.5481** | **0.5759** | **0.7357** |

Table 5: Ablation studies on refined training process. The best results are marked in **bold**.

| Exp | Condi Image | RAM | RealLR200 | | | DrealSR | | | |
|---|---|---|---|---|---|---|---|---|---|
| | | | MANIQA* ↑ | MUSIQ* ↑ | CLIPIQA* ↑ | MANIQA* ↑ | MUSIQ* ↑ | CLIPIQA* ↑ | PSNR↑ |
| (1) | ✗ | ✗ | 0.5623 | 70.75 | 0.7431 | 0.5331 | 64.55 | 0.7285 | **26.96** |
| (2) | ✓ | ✗ | 0.6092 | 71.76 | 0.7660 | 0.5372 | 62.67 | 0.6997 | 26.78 |
| (3) | ✗ | ✓ | 0.5772 | 71.33 | 0.7549 | 0.5433 | 65.09 | 0.7087 | 26.87 |
| AddSR | ✓ | ✓ | **0.6182** | **72.62** | **0.7724** | **0.6034** | **68.16** | **0.7381** | 26.09 |

Table 6: Ablation studies on PSR and TA-ADD.

| Methods | Time[s] | RealLR200 | | | DrealSR | | | |
|---|---|---|---|---|---|---|---|---|
| | | MANIQA* ↑ | MUSIQ* ↑ | CLIPIQA* ↑ | MANIQA* ↑ | MUSIQ* ↑ | CLIPIQA* ↑ | PSNR↑ |
| w/o PSR | 0.44∼0.77 | 0.5910 | 71.28 | 0.7541 | 0.5672 | 66.67 | 0.6589 | 25.85 |
| w/o TA-ADD | 0.44∼0.80 | 0.6058 | 72.19 | 0.7630 | 0.5898 | 67.58 | 0.7042 | 25.77 |
| AddSR | 0.44∼0.80 | **0.6182** | **72.62** | **0.7724** | **0.6034** | **68.16** | **0.7381** | **26.09** |

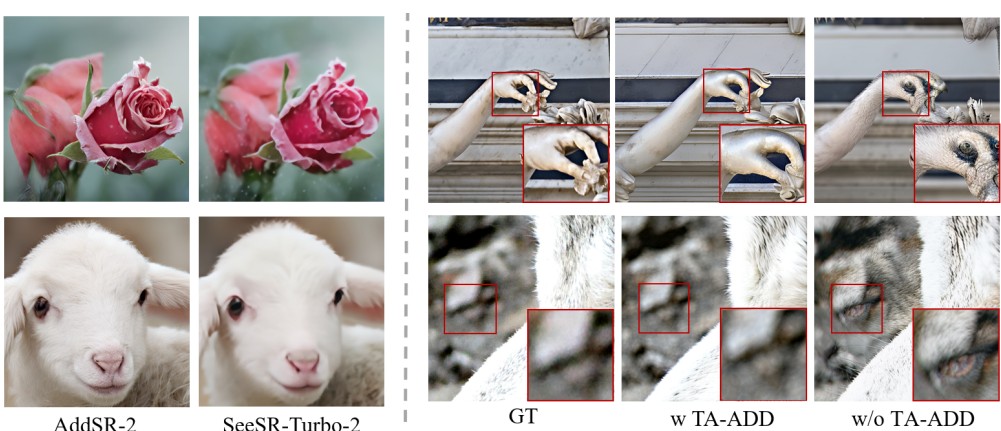

|  AddSR-2  |  SeeSR-Turbo-2  |  GT  |  w TA-ADD  |  w/o TA-ADD  |

Figure 11: **Left**: Comparison against SeeSR-Turbo-2. **Right**: Visual comparison of TA-ADD.

should appear blurry, while the RAM prompt renders the tree branches sharply. Conversely, the manual prompt maintains the background's intended blur, aligning with the Ground Truth.

**Comparison with the Efficient SeeSR-Turbo.** A recent efficient SD-based method named SeeSR-Turbo (19) has been introduced for blind super-resolution through 2-steps inference. To demonstrate the superiority of our AddSR, we conduct a visual comparison between SeeSR-Turbo and AddSR. The qualitative results are shown in Fig. 11-Left. One can see that our AddSR generates realistic textures by 2 steps, while SeeSR-Turbo tends to generate blurry results. We also provide quantitative comparison on RealLR200 and DIV2K on Tab. 4, Our AddSR surpasses SeeSR-Turbo across all displayed metrics, including PSNR, SSIM, NIQE, MANIQA, MUSIQ and CLIPIQA.

## 4.4 ABLATION STUDY

**Effectiveness of Refined Training Process.** To enrich the information provided by the teacher model, we refine the training process by substituting the LR image with HR image as inputs of ControlNet, RAM, and CLIP. Since SeeSR is adopted as the baseline, we also replace the LR image of its RAM input with HR image. The quantitative results are shown in Tab. 5. We can see that with the supervision from HR input, the perception quality of restored images becomes better.

**Effectiveness of TA-ADD on Balancing Perception and Fidelity.** The proposed TA-ADD aims to balance perception and fidelity quality of restored images. The quantitative results are shown in Tab. 6.

Table 7: Quantitative comparison of different settings for TA-ADD on synthetic degraded DIV2K. The best results are **bold**, and the second best results are underlined.

| Metrics | Real-ESRGAN | SeeSR | Ours-perception | Ours-fidelity |
|---|---|---|---|---|
| MANIQA↑ | 0.3374 | 0.5266 | **0.6335** | 0.5759 |
| CLIPIQ↑ | 0.5047 | 0.7180 | **0.7703** | 0.7357 |
| PSNR↑ | **22.70** | 21.86 | 21.45 | 21.92 |
| SSIM↑ | **0.5935** | 0.5474 | 0.5210 | 0.5481 |

Despite we increase the weight of the distillation loss in the later inference steps, the perceptual quality still improves. This could be attributed to the initial steps producing sufficiently high perception quality images, which offer more informative cues when combined with PSR. Consequently, the later inference steps can achieve high perceptual quality.

In addition, we can adjust the hyperparameters in TA-ADD and the number of inference steps to achieve competitive PSNR and SSIM results while excelling in perceptual quality. We primarily compare the leading methods in terms of fidelity (GAN-based Real-ESRGAN) and perceptual quality (SeeSR). As shown in Tab. 7, Our method remarkably enhances perceptual quality compared to SeeSR while also offering better fidelity. When compared to Real-ESRGAN, our method shows a substantial improvement in perceptual quality while maintaining comparable fidelity. This indicates that TA-ADD effectively navigates the perception-fidelity trade-off. Specifically, we made the following adjustments: 1) larger values for $\mu$ and $\nu$ ($\mu$=0.7, $\nu$=2.1) in TA-ADD during training, and 2) fewer inference steps (2 steps) to achieve high-fidelity results.

The visual results are shown in Fig. 11-Right. For the upper 3 images, the content is a statue. However, without TA-ADD, the model hallucinates its hand as a bird. For the bottom 3 images, the original background is rock. Again, without utilizing TA-ADD, AddSR might hallucinate the background as an eye of a wolf. Conversely, with the help of TA-ADD, the restored images can generate more consistent contents with GTs. TA-ADD constrains the model from excessively leveraging its generative capabilities, thereby preserving more information in the image content, aligning closely with the GTs. Specifically, using TA-ADD, texture of the statue's hand in the upper image remains unchanged, and the background of the bottom image retains the rock with out-of-focus appearance.

**Effectiveness of PSR.** As shown in Tab. 6, incorporating PSR significantly enhances perceptual quality with minimal computational cost. All of the three perception metrics, including MANIQA, MUSIQ and CLIPIQA, are improved on the two popular real-world datasets.

## 5 CONCLUSION

We propose AddSR, an effective and efficient model based on Stable Diffusion prior for blind super-resolution. To address the perception-distortion imbalance issue introduced by the original ADD, we introduce timestep-adaptive ADD, which assigns distinct weights to GAN loss and distillation loss across different student timesteps. In contrast to current SD-based BSR approaches that either use LR images to regulate each inference step's output or rely on additional modules to pre-clean LR images as conditions, AddSR substitutes the LR image with the HR image estimated in the preceding step. This substitution provides more high-frequency information, allowing for restored results with enhanced textures and edges, while maintaining efficiency. Additionally, we use the HR image as the controlling signal for the teacher model, enabling it to provide better supervision to the student model. Extensive experiments demonstrate that AddSR can generate superior results within 1∼4 steps in various degradation scenarios and real-world low-quality images.

**Limitations.** Although the inference speed of our AddSR surpasses all of the existing SD-based methods remarkably, there still exists a gap between AddSR and GAN-based methods. The primary factor is that AddSR is built upon SD and ControlNet, which, due to its substantial model parameters and intricate network structure, noticeably hinders the inference time. In the future, we plan to explore a more streamlined network architecture to boost overall efficiency.

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

# Appendix

## A  COMPARISONS AMONG ADD, SeeSR AND AddSR

In this section, we provide a comparison among ADD, SeeSR, and our proposed AddSR. Their architecture diagrams are depicted in Fig. 12. Firstly, the distinctions between ADD and AddSR primarily lie in two aspects: 1) *Introduction of ControlNet*: ADD is originally developed for text-to-image task, which typically only takes text as input. In contrast, AddSR is an image-to-image model that requires the additional ControlNet to receive information from the LR image. 2) *Perception-distortion Trade-off*: ADD aims to generate photo-realistic images from texts. However, introducing ADD into blind SR brings the perception-distortion imbalance issue (please refer to Sec. 3.4 in our submission), which is addressed by our proposed timestep-adaptive ADD in AddSR.

Secondly, the key differences between SeeSR and AddSR are: 1) *Introduction of Distillation*: SeeSR is trained based on vanilla SD model that needs 50 inference steps, while AddSR utilizes a teacher model to distill an efficient student model to achieve just 1∼4 steps. 2) *High-frequency Information*: SeeSR uses the LR image $y$ as the input of the ControlNet. In contrast, AddSR on one hand adopts the HR image $x_0$ as the input of the teacher model's ControlNet to supply the high-frequency signals since the teacher model is not required during inference. On the other hand, AddSR proposes a novel prediction-based self-refinement (PSR) to further provide high-frequency information by replacing the LR image with the predicted image as the input of the student model's ControlNet. Therefore, AddSR has the ability to generate results with more realistic details.

## B  EFFECTIVENESS OF PREDICTION-BASED SELF-REFINEMENT

Our PSR is proposed to remove the impact of LR degradation and enhance high-frequency signals to regulate the student model output. As shown in Fig. 13, the restored images generated with PSR exhibit more details and sharper edges, while the images generated without PSR tend to be blurry with fewer details.

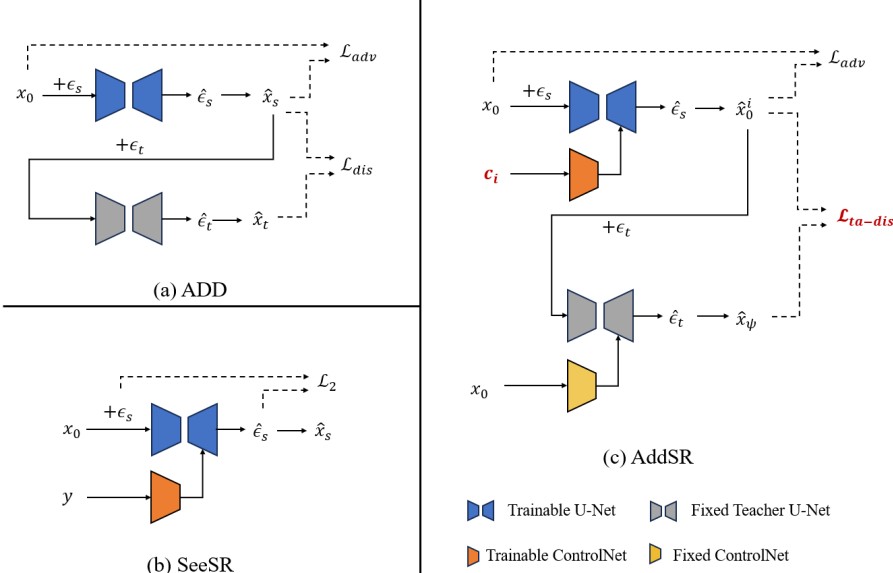

Figure 12: Comparisons on architecture diagram among ADD, SeeSR and AddSR. $x_0$ and y denote HR and LR images, respectively. $\hat{x}_0^i$ and $\hat{x}_\phi$ denote the predicted $x_0$ from the timesteps $s$ and $t$, respectively. $\epsilon_s$, $\epsilon_t$, $\hat{\epsilon}_s$ and $\hat{\epsilon}_s$ stand for added and predicted noise in timesteps $s$ and $t$, respectively. $\mathcal{L}_{ta-dis}$ is the timestep-adaptive distillation loss.

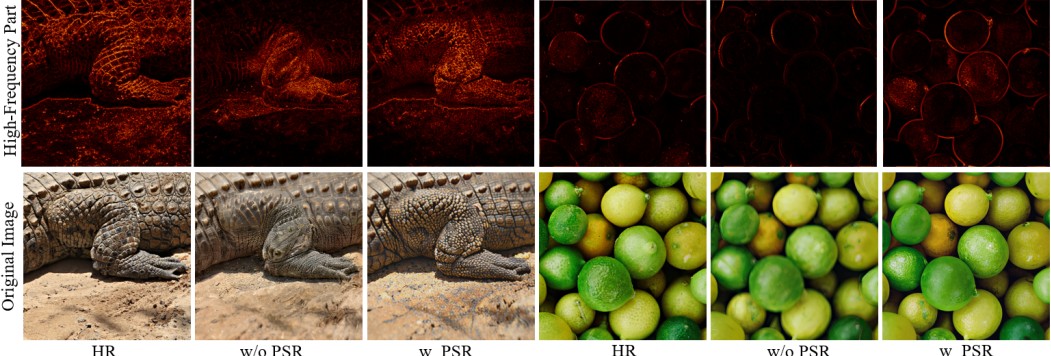

Figure 13: Our PSR is able to enhance the high-frequency signals of restored images to generate more photo-realistic details. The high frequency part is obtained using Fourier transform and filtering. Please zoom in for a better view.

## C  COMPARISON OF TIMESTEP-ADAPTIVE ADD FORMS

To determine the optimal settings for timestep-adaptive ADD, we conduct experiments on its different forms: exponential and linear. Specifically, the exponential form is defined as Eq. 3, while the linear form is defined as follows:

$$d(s,t) = (\prod_{i=0}^{t}(1-\beta_t))^{\frac{1}{2}} \times (\gamma \cdot p(s) + \kappa) \tag{4}$$

where the hyper-parameter $\kappa$ sets the initial weighting ratio, while $\gamma$ controls the increase of distillation loss over student timesteps. The quantitative results of the exponential and linear forms under various settings are listed in Tab. 8 and Tab. 9, respectively. The best settings for different forms in the tables are highlighted with a gray background. From these tables, we can draw the following conclusions: (1) The best results of the exponential form are better than those of the linear form. Therefore, we use Eq. (3) as the distillation loss function. Moreover, when $\mu = 0.5$ and $\nu = 2.1$, we achieves

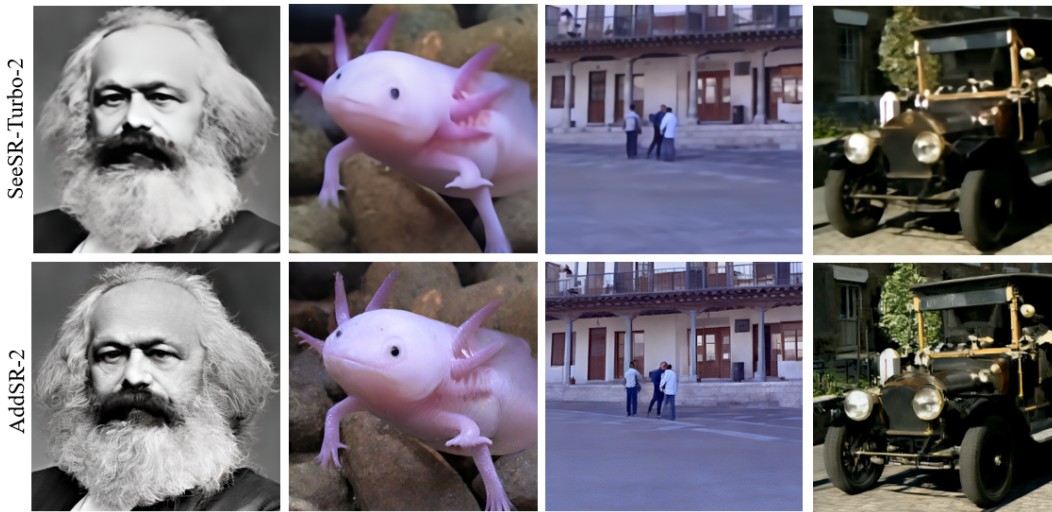

Figure 14: Visual comparisons between SeeSR-Turbo and AddSR. All the results are generate by 2 steps. Please zoom in for a better view.

Table 8: Comparing the exponential form of timestep-adaptive ADD across different hyper-parameters.

| $\mu$ | $\nu$ | \multicolumn{8}{c}{RealSR} | | | | | | | |
| | | \multicolumn{2}{c}{1 step} | | \multicolumn{2}{c}{2 step} | | \multicolumn{2}{c}{3 step} | | \multicolumn{2}{c}{4 step} | |
| | | MANIQA↑ | PSNR↑ | MANIQA↑ | PSNR↑ | MANIQA↑ | PSNR↑ | MANIQA↑ | PSNR↑ |
|---|---|---|---|---|---|---|---|---|---|
| | 1.3 | 0.4202 | 24.11 | 0.6263 | 23.10 | 0.6427 | 22.36 | 0.6453 | 22.33 |
| 0.5 | 1.7 | 0.3986 | 24.18 | 0.6110 | 24.01 | 0.6195 | 23.44 | 0.6307 | 23.40 |
| | 2.1 | 0.4189 | 24.22 | 0.6339 | 23.29 | 0.6496 | 22.76 | 0.6597 | 22.73 |
| | 2.5 | 0.4207 | 24.48 | 0.5939 | 23.92 | 0.6081 | 23.33 | 0.6197 | 23.29 |
| 0.7 | 2.1 | 0.3821 | 24.90 | 0.5971 | 24.03 | 0.6078 | 23.39 | 0.6221 | 23.36 |
| 0.9 | | 0.4095 | 23.16 | 0.6052 | 22.97 | 0.6062 | 22.88 | 0.6244 | 22.68 |

the best perceptual quality while maintaining good fidelity, so we use this setting for Eq. (3). (2) Increasing the hyper-parameters that control the distillation loss ratio (i.e., $\nu$ and $\gamma$) typically results in higher fidelity. For instance, when we fix $\mu$ to 0.5 and increase $\nu$, the overall trend in 4 step shows a decrease in perception quality and an improvement in fidelity. Consequently, we can achieve a perception-distortion trade-off by adjusting $\nu$.

Table 9: Comparing the linear form of timestep-adaptive ADD across different hyper-parameters.

| $\gamma$ | $\kappa$ | RealSR | | | | | | | |
|---|---|---|---|---|---|---|---|---|---|
| | | 1 step | | 2 step | | 3 step | | 4 step | |
| | | MANIQA↑ | PSNR↑ | MANIQA↑ | PSNR↑ | MANIQA↑ | PSNR↑ | MANIQA↑ | PSNR↑ |
| 0.1 | 0.7 | 0.3908 | 24.28 | 0.5849 | 23.64 | 0.6133 | 23.00 | 0.6172 | 22.98 |
| 0.2 | 0.3 | 0.4574 | 23.69 | 0.6294 | 23.06 | 0.6465 | 22.48 | 0.6480 | 22.41 |
| | 0.5 | 0.4338 | 23.87 | 0.6237 | 23.17 | 0.6407 | 22.55 | 0.6443 | 22.52 |
| | 0.7 | 0.4225 | 24.02 | 0.6064 | 23.24 | 0.6313 | 22.60 | 0.6352 | 22.59 |
| 0.4 | 0.1 | 0.4027 | 24.41 | 0.6077 | 23.25 | 0.6157 | 22.51 | 0.6215 | 22.45 |
| | 0.3 | 0.4045 | 24.39 | 0.5981 | 23.30 | 0.6124 | 22.58 | 0.6202 | 22.51 |
| | 0.5 | 0.4152 | 24.73 | 0.6261 | 23.33 | 0.6495 | 22.65 | 0.6507 | 22.62 |
| 0.6 | 0.1 | 0.4156 | 24.48 | 0.6175 | 23.44 | 0.6261 | 22.82 | 0.6328 | 22.79 |
| | 0.3 | 0.3926 | 24.90 | 0.5905 | 23.93 | 0.5857 | 23.51 | 0.5981 | 23.42 |
| 0.8 | 0.1 | 0.3887 | 24.92 | 0.5943 | 23.84 | 0.6038 | 23.36 | 0.6130 | 23.28 |

