# OpenReview forum: "AddSR: Accelerating Diffusion-based Blind Super-Resolution with Adversarial Diffusion Distillation"
_ICLR.cc/2025/Conference — ICLR 2025 Conference Withdrawn Submission_

### Official Review · Reviewer_aDzM · 2024-10-26

**Soundness:** 3
**Presentation:** 3
**Contribution:** 3
**Rating:** 6
**Confidence:** 5

**Summary:**

This paper introduces a model named AddSR, which addresses the challenge of blind super-resolution by leveraging the capabilities of stable diffusion. The authors propose prediction-based self-refinement and adversarial diffusion distillation methods to optimize the model, resulting in significantly improved efficiency and image quality. AddSR demonstrates superior performance on various degradation scenarios and real-world low-quality images, showcasing its effectiveness on tasks such as image restoration within a remarkably reduced number of inference steps.

**Strengths:**

- The paper introduces AddSR, a model that significantly advances blind super-resolution by employing stable diffusion, showcasing impressive generative capabilities for reconstructing high-resolution images from low-resolution inputs.
- Two innovative aspects of the paper are the prediction-based self-refinement strategy, which efficiently incorporates high-frequency details, and the adversarial diffusion distillation approach, which accelerates the model's inference speed while maintaining quality.
- The authors have meticulously designed experiments that thoroughly evaluate AddSR's performance across a variety of datasets, demonstrating the model's robustness and effectiveness in different degradation scenarios.
- The writing is clear and well-structured, making the complex technical details accessible and the methodology easy to follow, which is commendable.

**Weaknesses:**

- Could you point out the extract reasons about that directly applying ADD in the BSR task leads to reduced fidelity? The description in this paper is confusing, only through the observation of experimental results, lack of theoretical analysis.

- The latest diffusion-based super resolution methods[1,2,3,4] have accelerated the inference process into single step, but this paper only obtains the optimal results with 4-steps,  It seems that the method in this paper does not have advantages and practicality. could you compare with them?

- PSR sounds insteresting, but when the estimated HR images are rough and distorted in the bigger steps, PSR will transfer the distortion error to the next step, LR is necessary, How to reduce the error?

- Time-step aware weighting is usefully, but the functions d(s, t) is too empirical, the times-step in TAD-SR[2] shows more simpler and more effective, similar to diffusion time-step embedding.

[1] Wu, Rongyuan, et al. "One-Step Effective Diffusion Network for Real-World Image Super-Resolution." arXiv preprint arXiv:2406.08177 (2024).

[2] He, Xiao, et al. "One Step Diffusion-based Super-Resolution with Time-Aware Distillation." arXiv preprint arXiv:2408.07476 (2024).

[3] Noroozi, Mehdi, et al. "You Only Need One Step: Fast Super-Resolution with Stable Diffusion via Scale Distillation." arXiv preprint arXiv:2401.17258 (2024).

[4] Zhang, Aiping, et al. "Degradation-Guided One-Step Image Super-Resolution with Diffusion Priors." arXiv preprint arXiv:2409.17058 (2024).

**Questions:**

Please referring to the weaknesses above.

---

### Official Review · Reviewer_hSbP · 2024-10-31

**Soundness:** 3
**Presentation:** 4
**Contribution:** 2
**Rating:** 5
**Confidence:** 4

**Summary:**

Previous diffusion-based SR methods face challenges related to perception distortion imbalance and sensitivity to the quality of conditioning inputs. To address these issues, this paper introduces TA-ADD and a self-refinement strategy, respectively. Specifically, instead of using the same LR image as a condition across timesteps, the self-refinement strategy utilizes the predicted \hat{x_{0}}​ from previous steps, offering richer information for subsequent denoising. Additionally, to manage the imbalanced loss weights between adversarial and distillation losses at different timesteps, the authors propose a weighting function to adjust the loss weights accordingly.

**Strengths:**

1. The proposed methodologies appear straightforward yet powerful. In comparison to previous SR methods, AddSR demonstrates superior performance in both qualitative and quantitative analyses.

**Weaknesses:**

1. While the primary focus of the paper is on PSR and TA-ADD, there are significant concerns regarding its novelty. First, although several works utilize the output from previous steps as a condition for the next denoising process*, what distinguishes the proposed TA-ADD from *? Second, since ** also employs a weighting function to adaptively adjust the loss term weights, what advantages does Equation 3 offer compared to their approach? If possible, it would be helpful to include a comparison using their weighting functions.

*Andreas Lugmayr, RePaint: Inpainting using Denoising Diffusion Probabilistic Models

**Tianwei Yin, One-step Diffusion with Distribution Matching Distillation

**Axel Sauer, Fast High-Resolution Image Synthesis with Latent Adversarial Diffusion Distillation

2. What advantages does AddSR-1 gain from self-refinement? Additionally, please provide a diverse comparison with other one-step diffusion methods* in Table 1 for a more comprehensive understanding.

*Y Wang, SinSR: Diffusion-Based Image Super-Resolution in a Single Step

*R Wu, One-Step Effective Diffusion Network for Real-World Image Super-Resolution

*M Noroozi, You Only Need One Step: Fast Super-Resolution with Stable Diffusion via Scale Distillation

3. While this is not a weakness, the reviewer suggests that the authors align the abstracts in both the OpenReview submission and the paper. The abstract on OpenReview seems to be the earlier version before revisions were made.

**Questions:**

1. Let the reviewer denote Figure 4's \hat{x}^{1}_{0}...\hat{x}^{4}_{0} to 4-1, 4-2, 4-3, and 4-4. In Figure 4, it appears that 4-1 is more similar to 4-4 than to 4-2. Since 4-1 is generated using 4-2 as a condition, one would expect it to be more similar to 4-2. What could explain this discrepancy?

2. Given that PSR can be applied to previous stable diffusion-based SR models without the need for retraining, could the authors present both quantitative and qualitative results showing the performance of PSR on other models?

3. Instead of using enhanced conditions in PSR, would it be possible to substitute all LR images with predicted SR images from simpler SR methods like RealESRGAN? If the richer conditional information from PSR is a crucial factor, then using a naively super-resolved image could be more beneficial during the initial timesteps. Here, instead of retraining the model with SR image conditions, it might suffice to replace the \hat{x}^{0}​ condition with outputs from RealESRGAN on a pretrained AddSR model.

---

### Official Review · Reviewer_jczQ · 2024-11-04

**Soundness:** 3
**Presentation:** 3
**Contribution:** 2
**Rating:** 5
**Confidence:** 4

**Summary:**

This paper presents AddSR, a blind super-resolution method using Timestep-Adaptive ADD (TA-ADD) to address challenges of perception-distortion imbalance and high-frequency detail restoration. AddSR achieves high-quality image restoration more efficiently, demonstrating a 7x speed advantage over existing models in experiments.

**Strengths:**

1. The task is meaningful and engaging.

2. Achieves good performance on certain non-reference metrics.

3. High efficiency compared to SeeSR.

**Weaknesses:**

Performance Drop: There is a noticeable performance gap between the teacher and student models, with LPIPS dropping from 0.2124 to 0.2953—a significant decrease. From my experience, LPIPS is a more crucial metric than the non-reference metrics, where the proposed method shows improvement.

Notation Error: The notation is incorrect. In L271-272, the authors state, "'*' indicates that the metric is non-reference," and label LPIPS as non-reference, although it is actually reference-based.

Missing LPIPS and Comparison: LPIPS is absent from Table 3, and the fidelity performance appears much lower than existing SOTA models, such as ResShift, SeeSR, and even current single-step SR models. Comparisons with these methods are also missing.

Complexity and Missing Details: The proposed method seems somewhat complex, and certain details are unclear. For instance, the caption for Fig. 2 lacks essential information, details on PSR are not provided, and the input for Fig. 3 is unspecified.

**Questions:**

See the weekness part.

---

### Official Review · Reviewer_hXqt · 2024-11-08

**Soundness:** 2
**Presentation:** 2
**Contribution:** 1
**Rating:** 3
**Confidence:** 4

**Summary:**

This work focuses on enhancing the efficiency of SD-based BSR methods by incorporating the Adversarial Diffusion Distillation (ADD) technique. Additionally, this study points out the perception-fidelity imbalance issue and the impact of image condition when applying ADD, proposing TA-ADD loss and prediction-based self-refinement (PSR) mechanism to address them, respectively.

**Strengths:**

1. Some qualitative results demonstrate high-quality details.
2. The leading results in perceptual-oriented metrics.

**Weaknesses:**

1. The main focus of this work is to emphasize acceleration through ADD, but most of the discussions revolve around addressing the limitations of ADD, then introducing certain tricks (i.e., loss function, improving image condition) to tackle the problems, without demonstrating noticeable effects in improving efficiency.

2. Another focus of this paper is to resolve the perception-distortion trade-off using the proposed TA-ADD loss. However, from Tables 1 and 3, it is evident that this framework fails to push the boundary of this trade-off. Instead, this framework achieves improved perceptual quality by heavily sacrificing fidelity, as indicated by low PSNR/SSIM scores.

3. The lack of comparison with other efficient SD-based methods (e.g., OSEDiff) makes it difficult to verify whether the effectiveness of this framework under similar efficiency.

4. On main focus of this work is to address perception-fidelity imbalance. In practice, SD-based SR methods can trade off between fidelity perceptual quality by employing inference tricks (e.g., manually added text prompts). However, this paper does not specify the inference setup of each baseline, nor does it achieve promising results with an acceptable level of fidelity drop.

5. In Table 1, the performance of this framework is not comparable with existing SD-based methods in several metrics. Specifically, while the proposed method shows better visual quality, it significantly underperforms in fidelity-oriented metrics such as PSNR and SSIM.

6. In Table 2, this framework doesn't deliver similar perceptual quality scores as SUPIR, such that comparing other settings at this point does not adequately demonstrate the advantages of this approach.

7. Confusing terminology. For example, in the caption of Figure 3, the term "predicted HR" image is used. Typically, "HR image" refers to ground truth. Since your method also uses the ground truth HR image as a condition for the teacher model, mixing the use of the term "HR" might lead to misunderstanding.

**Questions:**

1. This paper emphasizes that the proposed PSR can utilize the HR output to "control" the final output, but the experiment about how to control the intermediate results and influence the final output is missing.

2. PSR leverages the intermediate SR result as the condition for the next iteration. However, as shown in Figure 5(b), SD-based SR methods often have hallucination issues. Did you consider error accumulation in this scenario? Did you apply any additional processing to the intermediate output (e.g., blurring) to prevent error accumulation?

3. Line 230: "The $\hat{x_0}$ in each step has more high-frequency information to better control the model output." However, in Figure 5(b), the hallucinated animal head also contains high-frequency components. How do you prove that high-frequency components necessarily provide better guidance?

4. This work adopts SeeSR as the backbone and teacher model. Did you use manually added prompts when generating images, such as "clean," "high-resolution," or "8k" as used in SeeSR? Since these prompts greatly influence the perception-fidelity trade-off in SR output, their use may also affect your model's performance when intermediate results are regarded as crucial conditions. Could you compare the impact on performance with and without these prompts?

5. Why does your method, which uses the same framework as SeeSR and reduces the number of steps from 50 to 4, only achieve a 7$\times$ speedup (which ideally should be 12.5$\times$)?

6. The entire training scheme appears to be overfitting to the 4-step setting, however, the ablation about the choice of this hyperparameter is missing. What will happen if you use the trained student model for 5 or more steps during inference?

7. In the design of the $L_{ta-dis}$ loss, why do you use the output of the student model as the input of the teacher model, then enforcing similarity between the teacher output and student output? This implies that the purpose of the $L_{ta-dis}$ is for the teacher to learn identity rather than to improve the student. However, since the teacher does not have any learnable layers, this design cannot derive such effect.

---

### Official Review · Reviewer_dfRH · 2024-11-13

**Soundness:** 3
**Presentation:** 2
**Contribution:** 3
**Rating:** 6
**Confidence:** 1

**Summary:**

This paper presents AddSR, an efficient blind image super-resolution method based on Adversarial Diffusion Distillation (ADD). It introduces a timestep-adaptive adversarial diffusion distillation (TA-ADD) loss to address the perception-distortion imbalance inherent in ADD-based approaches. Additionally, it proposes a prediction-based self-refinement strategy to improve high-resolution (HR) estimation.

**Strengths:**

1. PSR does not use additional modules to pre-clean LR images. Instead, it uses HR images estimated by Equation (2).

2. AddSR’s teacher model substitutes $x_{LR}$ with $x_0$ to regulate the output, which could force the student model to learn the high-frequency information of HR images implicitly.

3. AddSR can produce images with better perceptual quality than most diffusion-based SR models, requiring fewer inference steps and less time.

**Weaknesses:**

1. The texts in Figure 6 and Table 1 are too small.

2. This paper does not provide the type of GPUs, and inference time when testing.

3. Lack of visual comparisons with SUPIR[1].

4. The visual comparisons are not thorough.


[1] Gu Jinjin, et  al. Pipal: a large-scale image quality assessment dataset for perceptual image restoration.

**Questions:**

1. The texts in Figure 6 and Table 1 are too small. Please reformat them.

2. What type of GPU are the testing experiments done?

3. Could you provide the visual comparison results with SUPIR [1]?

4. Could you provide the inference times when comparing with SotAs in Table 1?

5. Please provide more visual comparisons with SotA methods as supplements.

6. How to use HR images as conditions when training prediction-based self-refinement? What are the differences between training and inference when using HR as a condition?

7. Since there is no HR input during inference, what is the significance of the *Effectiveness of Refined Training Process* in the ablation study?

8. Does the ratio of GAN loss to distillation loss vary with different scenarios? If yes, how to adjust the ratio in different scenarios? Do we need to change the ratio for each image?

9. Please describe the effects of using only GAN loss and only distillation loss.

[1] Gu Jinjin, et  al. Pipal: a
large-scale image quality assessment dataset for perceptual image restoration.

---

### Author Response · Authors · 2024-11-27

Thanks for the time of ACs and reviewers, we decide to withdrawal our paper.

---

### Note · Authors · 2024-11-27

I have read and agree with the venue's withdrawal policy on behalf of myself and my co-authors.